# Dreaming is All You Need

## Abstract

Achieving a harmonious balance between exploration and precision is of paramount importance in classification tasks. To this end, this research introduces two novel deep learning models, SleepNet and DreamNet, to strike this balance. SleepNet seamlessly integrates supervised learning with unsupervised "sleep" stages using pre-trained encoder models. Dedicated neurons within SleepNet are embedded in these unsupervised features, forming intermittent "sleep" blocks that facilitate exploratory learning. Building upon the foundation of SleepNet, DreamNet employs full encoder-decoder frameworks to reconstruct the hidden states, mimicking the human "dreamin" process. This reconstruction process enables further exploration and refinement of the learned representations. Moreover, the principle ideas of our SleepNet and DreamNet are generic and can be applied to both computer vision and natural language processing downstream tasks. Through extensive empirical evaluations on diverse image and text datasets, SleepNet and DreanNet have demonstrated superior performance compared to state-of-the-art models, showcasing the strengths of unsupervised exploration and supervised precision afforded by our innovative approaches.

## 1 Introduction

In the current digital age, the ability to accurately classify large datasets has become of paramount importance across a myriad of fields, including computer vision (CV) Krizhevsky et al. (2012); Simonyan & Zisserman (2014); Szegedy et al. (2014); Dosovitskiy et al. (2020a), natural language processing (NLP) Mikolov et al. (2013); Peters et al. (2018); Devlin et al. (2019); Brown et al. (2020), bioinformatics Jumper et al. (2021), etc. The blossoming of artificial intelligence and deep learning has greatly facilitated the handling of complex classification tasks. Deep learning's capacity to sift through multitudes of variables, discern patterns, and extract key features has led to impressive breakthroughs in numerous applications, from image recognition and voice recognition to disease prediction Ilyas et al. (2019).

The groundbreaking convolutional neural networks (ConvNets), such as ResNetHe et al. (2015) and EfficientNetTan & Le (2019), have emerged as dominant architectures in computer vision, with ResNet addressing the vanishing gradient issue through deep residual networks and enabling deeper models without performance loss, while EfficientNet introduced a compound scaling method that scales depth, width, and resolution, enhancing both efficiency and accuracy. These models have set new benchmarks across various datasets and have been pivotal in applications such as autonomous driving and advanced image recognition, reshaping how machines interpret visual data. Meanwhile, the success of pre-trained unsupervised Transformers Vaswani et al. (2017) like ViT Dosovitskiy et al. (2020a) for vision tasks and BERT Devlin et al. (2019) has shown that using primarily standard Transformer layers can achieve significant performance in downstream applications, reaching levels comparable to previous state-of-the-art neural networks and suggesting that Transformers may offer greater scalability across diverse domains.

Transformers have demonstrated superior model capabilities but often suffer from poor generalization when compared to chain-like networks due to a lack of appropriate inductive bias Wu et al. (2021). Recent research has focused on hybrid methods that combine the structures of both to retain their respective advantages Dosovitskiy et al. (2020b); Wu et al. (2021); Dai et al. (2021); Huang et al. (2020). For example, Convolution Vision Transformers (CvT) Wu et al. (2021) enhance performance by integrating convolutional token embedding and convolutional transformer blocks with convolutional projection, aiming for improved

accuracy and efficiency. Similarly, the convolution and attention transformer (CoAtNet) Dai et al. (2021) boosts performance by introducing relative attention that merges convolution and attention mechanisms, enhancing generalizability and efficiency through a simple stacking of convolution and attention layers. In the realm of natural language processing, the Transformer with BLSTM (TRANS-BLSTM) Huang et al. (2020) integrates a BLSTM layer into each transformer block, creating a joint modeling framework that leverages both transformer and BLSTM technologies.

However, these models still lack generalizability as their evaluations are often limited to specific transformers and focus primarily on testing the architecture superficially, without delving into deeper conceptual explorations, potentially leading to a lack of broad applicability. In response to this challenge, we propose two innovative networks designed for both vision and textual classification tasks. The first, SleepNet, introduces a revolutionary learning paradigm that incorporates "sleep" blocks into the training process of neural networks. Drawing inspiration from cognitive science, SleepNet integrates pre-trained features into designated neurons, creating "sleep" periods interspersed within supervised learning epochs. This process mirrors the role of sleep in human memory consolidation, akin to a symphony conductor orchestrating the integration of experiences. Notably, SleepNet preserves the weights of the pretrained encoding components in each Sleep connection during the supervised phases, imitating human memory consolidation during sleep. Building on this foundation, we developed DreamNet, which enhances SleepNet by introducing a "dream" block. Unlike SleepNet, DreamNet utilizes a complete unsupervised pre-trained autoencoder, not just the encoder, to deepen feature consolidation. This autoencoder not only reconstructs the hidden states but also serves as an additional feature enhancer compared to SleepNet, thereby boosting overall performance.

This research contributes significantly to the field of deep learning in the following crucial ways:

- We present an innovative deep learning strategy, **SleepNet**, which mimics memory consolidation during sleep to enhance deep learning processes. SleepNet combines pre-trained encoders with supervised learning in a hybrid framework, enabling rich data interpretation and robust supervised predictions.

- Building on SleepNet, we introduce **DreamNet**, which leverages a pre-trained encoder-decoder framework for feature augmentation. This approach enhances knowledge transfer and consolidates training, resulting in further performance improvements.

- Both SleepNet and DreamNet have significant potential for general applicability. We introduce two versions of models for both SleepNet and DreamNet, one for CV tasks and the other for NLP tasks, to show their general task capabilities. Extensive experiments demonstrate that the proposed methods, especially DreamNet, consistently outperform start-of-the-art baselines on both image and text classification tasks, highlighting their superior efficacy and potential for universal applicability.

The rest of this paper is structured as follows. We first review the existing deep learning methods in CV and NLP and provide a brief discussion of biological dreams in Section 2. Then we detail our proposed methods in Section 3. We evaluate the performance of our proposed method through empirical analysis in Section 4. We conclude the paper with suggestions for future work in Section 5.

## 2 Related Work

Deep learning has been significantly advanced with the development of supervised learning models like ResNet He et al. (2015), MobileNet Howard et al. (2017), and TextCNN Kim (2014), which excel in multiple tasks across image processing, object detection, and natural language processing. These models share a common architecture: chain-like structures where components are sequentially linked, facilitating a systematic enhancement of feature extraction. For instance, ResNet utilizes residual blocks connected by skip connections to tackle the vanishing gradient problem in deep networks, while MobileNet is optimized for mobile environments with streamlined CNN structures. TextCNN revolutionizes text classification by applying convolutional layers directly to text, capturing various n-gram features with global max-pooling. Although highly effective in recognizing patterns from vast labeled datasets, this reliance on extensive labeled data can limit their adaptability and generalization in new, less structured environments.

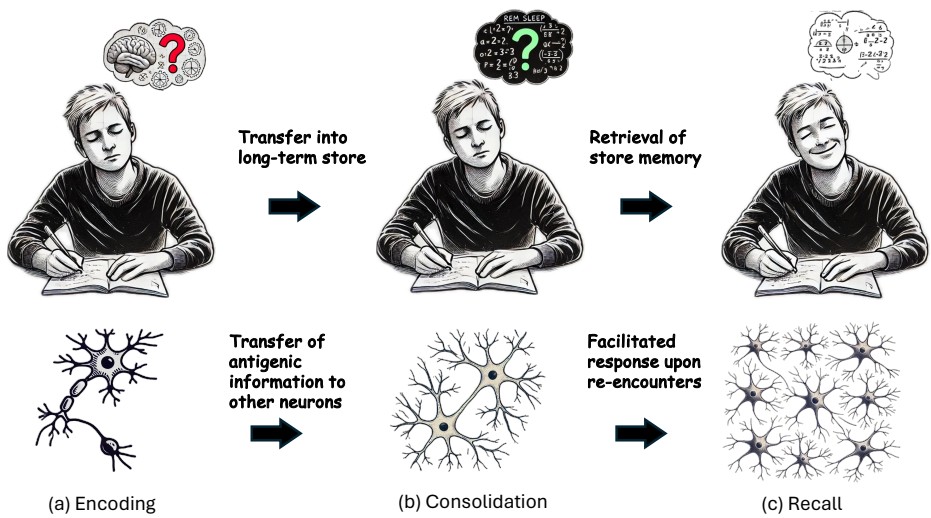

Figure 1: This diagram illustrates how sleep and dreams act as the foundation for memory formation. Sleep facilitates three key phases: encoding, where information is sensed; consolidation, where it is stored long-term; and recall, where it is retrieved Besedovsky et al. (2019). By fostering optimal conditions for neuronal interactions, sleep strengthens memory retention and recall Goldstein & Walker (2014).

Unsupervised learning models, such as Variational Autoencoders (VAEs) Kingma & Welling (2013) and Generative Adversarial Networks (GANs) Goodfellow et al. (2014), play a key role in understanding the inherent structure and distribution of data without explicit labels, with applications ranging from image generation to style transfer. However, they often fall short in task-specific applications like classification. In contrast, Transformers Vaswani et al. (2017), particularly with their self-attention capabilities, have revolutionized neural language processing and speech understanding. Notable implementations like BERT Devlin et al. (2019), which utilizes multi-head and self-attention, have set new benchmarks in processing long documents. Vision Transformer (ViT) Dosovitskiy et al. (2020a) extends this approach to image classification, achieving remarkable results after pre-training on the large-scale JFT dataset, although it struggles with limited data and generality Wu et al. (2021).

The integration of convolutional layers with unsupervised models, especially transformers, has led to several innovative hybrid architectures, blending the strengths of CNNs, RNNs, and transformers to enhance feature extraction and understanding of local relationships. Notable implementations include replacing multi-head attention with convolution layers or adding them sequentially or in parallel to transformer blocks. Specific examples like Convolution Vision Transformers (CvT) Wu et al. (2021) utilize convolutional projections instead of traditional position-wise linear projections for attention mechanisms, while convolution and attention transformer (CoAtNet) Dai et al. (2021) combines convolution and attention layers to improve performance and efficiency. Additionally, hierarchical multi-stage structures similar to CNNs have been introduced, significantly boosting performance. These methods augment traditional ConvNets with self-attention modules or incorporate convolutional properties directly into transformer backbones such as ResNet-ViT Dosovitskiy et al. (2020b), demonstrating the potential of these hybrid models to leverage the strengths of both paradigms effectively.

Deep learning seeks to improve efficiency and generalization by consolidating learned representations. Drawing from biological principles, researchers are actively exploring iterative feature refinement Jaderberg et al. (2017), knowledge transfer Hinton et al. (2015), and hierarchical representation learning Bengio et al. (2013). Drawing inspiration from representational consolidation observed in biological neural networks during sleep and rest states Rasch & Born (2013); Goldstein & Walker (2014), as illustrated in Fig 1, we explore analogous computational strategies for enhancing deep learning models. Specifically, we frame information transfer and reconstruction as feature augmentation and representation regularization opportunities. The following sec-

tions introduce methods that leverage these principles to improve task performance, reduce computational complexity, and enhance model robustness.

## 3 Methodology

In this section, we provide details of our proposed innovative deep learning architectures, SleepNet and DreamNet.

### 3.1 Problem Setting

In general, deep neural networks are constructed by connecting many weight matrices and nonlinear operators. In this paper, we consider a chain-like neural network constructed by stacking similar deep neural blocks, such as Multilayer Perceptron, stacked LSTMs and ResNet. Let $D = (x_1, y_1), (x_2, y_2), \ldots, (x_n, y_n)$ be a dataset consisting of $n$ samples, where $x$ and $y$ represent the input text and its corresponding class, respectively. Given a chain classifier with $M$ layers, $C(x) = (g_M \circ g_{M-1} \circ \ldots \circ g_2 \circ g_1)(x)$ maps from the text space $\mathcal{X}$ to the $k$ classes, where $g_m(\cdot)$ is the $m$th neural blocks. The output of the $m$th block is $h_m(x) = g_m(h_{m-1}(x))$, and the neural blocks can be similar blocks constructed by fully connected layers, CNN, LSTM, pooling and normalizing layers with the activation functions. To construct a SleepNet, we also need a pre-trained and self-supervised autoencoder model $P = \theta(\phi(x))$ where $\theta(\cdot)$ and $\phi(\cdot)$ are the encoder and decoder, respectively. The hidden state from the latent space is $a = \phi(x)$.

### 3.2 SleepNet

The proposed method, SleepNet, incorporates a pre-trained unsupervised encoder to feed input and hidden states, producing enriched encodings for subsequent layers. As we shall explain in this section, this fusion of information inside a single architecture innovatively borrows the concept of sleep functions in human cognition and applies it to machine learning tasks.

#### 3.2.1 Sleep Connection

The SleepNet is designed to harness the strengths of pre-trained encoder into supervised learning models, consolidating them into one cohesive method. Its unique attribute incorporates a self-supervised encoder $\phi(\cdot)$, pre-trained on an unlabeled dataset based on the autoencoder framework. The reason for only utilizing the encoder is that a well-trained encoder is expected to transform more features to the input Yao et al. (2021), and excluding the decoder will reduce the complexity of the proposed method. We contend that such an encoder setup amplifies feature extraction, easing the learning process.

SleepNet uses a pre-trained encoder $\phi$ to process hidden states $h_i$, forwarding encoded features to subsequent blocks through a mechanism that we call "sleep connection." This approach harnesses latent features for the supervised learning process. "Sleep blocks" interspersed within the model bridge the output of supervised blocks $h_m = g_m(h_{m-1})$ with encoded features $\phi(h_{m-1})$, enhancing the integration of pre-trained insights into the learning sequence. The "sleep block" $s(\cdot)$ is mathematically expressed as:

$$s_m(\mathbf{h}_m) = g_m(\mathbf{h}_{m-1}) + \phi(\mathbf{h}_{m-1}), \tag{1}$$

with $\mathbf{h}_{m-1}$ denoting the output from the $m$th sleep block. The supervised blocks capture local details, whereas encoder $\phi$ elevates local to non-local information, facilitating enhanced feature fusion.

We devise two versions of SleepNet, one for computer vision (CV) tasks and the other for natural language processing (NLP) tasks.

#### 3.2.2 SleepNet for CV models

In computer vision applications, SleepNet begins by processing an input image through convolutional ("Cov") layers to extract preliminary features, which are then normalized ("Norm") and activated through ReLU functions to prepare them for further processing. The core of SleepNet consists of sleep blocks, each consisting

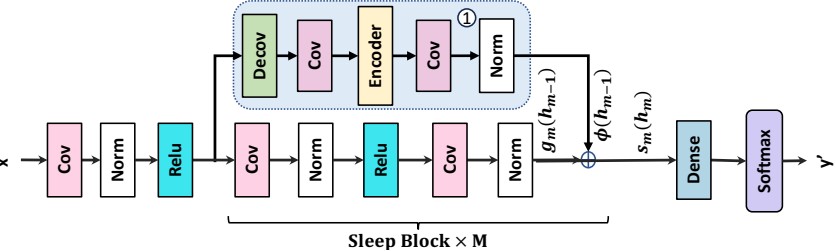

Figure 2: Overview of the Visual SleepNet Architecture, featuring M "Sleep Blocks" that are constructed by chain-like blocks processing data through convolutional layers ("Cov"), normalization ("Norm"), and sleep connection in block ①. Sleep connection includes an encoder and a deconvolution layer for feature extraction and dimension adjustment, mimicking cognitive sleep blocks. The workflow culminates in a dense layer followed by a softmax classifier for output classification.

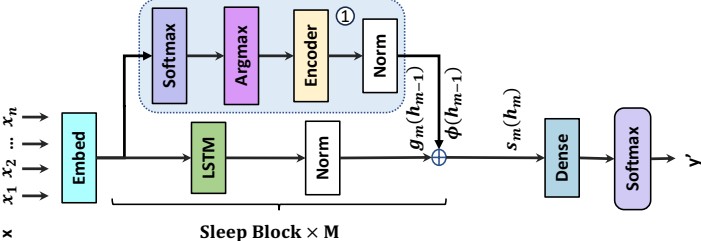

Figure 3: Overview of the Textual SleepNet Architecture, featuring M "Sleep Blocks" that are constructed by chain-like blocks processing data through LSTM, normalization ("Norm"), and sleep connection in block ①. Sleep connection includes softmax and argmax functions to make a legitimate sequence for the upcoming encoder for feature extraction, mimicking cognitive sleep blocks. The workflow culminates in a dense layer followed by a softmax classifier for output classification.

of a sleep connection and chain-like blocks. The sleep connection (Block ① in Fig 2) adjusts the feature dimensions via deconvolutional ("Decov") and convolutional layers to match the fixed input size required by the typically pre-trained encoder, $\phi(\cdot)$. This is essential because initial processing often reduces input dimensionality, potentially mismatching the encoder's specifications. After dimension adjustment and feature enhancement by the encoder, further convolutional and normalization layers refine and compress the features. Simultaneously, the initial output is processed through chain-like blocks for independent feature extraction. Outputs from both pathways are then merged by simple addition, repeating across multiple sleep blocks to progressively enhance features. This integrated workflow in Fig 2 culminates in a dense layer, leading to a softmax classifier for final image classification, effectively combining supervised and unsupervised learning to improve prediction accuracy.

### 3.2.3 SleepNet for NLP models

In natural language processing applications, SleepNet initiates its process by embedding input text using layers, such as ELMo Peters et al. (2018) or Word2Vec Mikolov et al. (2013), which transform raw texts into meaningful vector representations that capture semantic properties. Following embedding, the text progresses through chain-like blocks composed of LSTM units, adept at maintaining context over long sequences. Simultaneously, a sleep connection (Block ① in Fig 3) within each sleep block utilizes softmax and argmax functions to prepare sequences for the pre-trained encoder, $\phi(\cdot)$, enhancing feature representation. This encoder adjusts and enriches the LSTM-processed data, effectively integrating supervised learning from the LSTM blocks with unsupervised learning from the sleep connection. Outputs from both the sleep connection and LSTM blocks are merged through simple addition, repeating this integration across multiple blocks to refine text representation iteratively. The workflow in Fig 3 culminates in a dense layer followed

by a softmax classifier, which categorizes the text into predefined classes, leveraging both learning types to enhance the model's effectiveness in various NLP tasks.

### 3.3 DreamNet

Building on the SleepNet framework, DreamNet is designed to simulate and leverage dream-like processes during offline states. It enhances existing sleep connections with a novel "dream connection", utilizing a full autoencoder setup. Additionally, DreamNet learns augmented features by reconstructing hidden states using an autoencoder, akin to how humans draw inspiration from recalled dreams.

#### 3.3.1 Dream Connection

We propose a "dream connection" to enhance the previously proposed "sleep connection" through two innovative modifications. Firstly, we replace the encoder $\phi(.)$ with a complete pre-trained autoencoder $\theta(\phi(x))$. We anticipate that utilizing the full autoencoder will allow for a more comprehensive integration of information into the subsequent block. The mathematical illustration of a "dream connection" is as follows:

$$s_m(\mathbf{h}_m) = g_m(\mathbf{h}_{m-1}) + \theta(\phi(\mathbf{h_{m-1}})), \tag{2}$$

where $\theta(\cdot)$ and $\phi(\cdot)$ are encoder and decoder, respectively. Secondly, the output ("dream") generated by the autoencoder $\theta(\phi(x))$ is not only transferred to the subsequent block but also directed to a parallel block for a deeper analysis of these reconstructed features. In the final stage, the data from the 'dreams' and the chain-like module is combined before being processed through fully connected layers, ultimately leading to a softmax layer for predictions.

DreamNet is expected to achieve better performance by utilizing the pre-trained autoencoder for feature augmentation, which is theoretically and practically different from traditional data augmentation. Specifically, conventional augmentation methods, such as RandAugment Cubuk et al. (2020) and TextAttack Morris et al. (2020), typically involve crude manipulations on the raw dataset. While these methods can generate similar data, they may also introduce unnecessary noise. In contrast, feature augmentation through "dreaming" is a more principled approach that leverages the model's architecture. Practically, data augmentation and feature augmentation can be used complementarily within the same model, where data augmentation operates on the raw data, while our proposed feature augmentations with "dreams" are based on enhancing the model's internal representations.

Like SleepNet, we design two variations of DreamNets, one for CV tasks and the other for NLP tasks.

#### 3.3.2 DreamNet for CV Models

DreamNet enhances SleepNet's architecture for computer vision by incorporating a "dream connection" that employs a comprehensive autoencoder for advanced feature consolidation, as illustrated in Fig 4. Unlike SleepNet, DreamNet introduces "Dream Blocks" which not only include standard chain-like blocks for initial feature extraction but also feature an advanced dream connection setup (block ② in Fig 4). This dream connection uses a full autoencoder that first encodes the feature data to a latent, compressed representation and then decodes it, effectively reconstructing and enhancing the image data to simulate cognitive dreaming processes. This reconstructed output undergoes further refinement in block ③ in Fig 4, where additional convolutional and normalization layers refine these features to ensure robust feature extraction. This continuous cycle of encoding, decoding, and refinement significantly bolsters the model's pattern recognition capabilities. The workflow culminates in a dense layer followed by a softmax classifier, ensuring precise classification by effectively leveraging the enhanced feature set. To give a more valid illustration, we present Fig 6 by plotting the original image and the generated dream-like images.

#### 3.3.3 DreamNet for Textual Models

In its application to natural language processing tasks, DreamNet expands upon SleepNet by integrating a "dream connection" that utilizes a full autoencoder, as detailed in Fig 5. The process begins with the LSTM units, which initially process the text data to capture contextual dependencies. The output from these

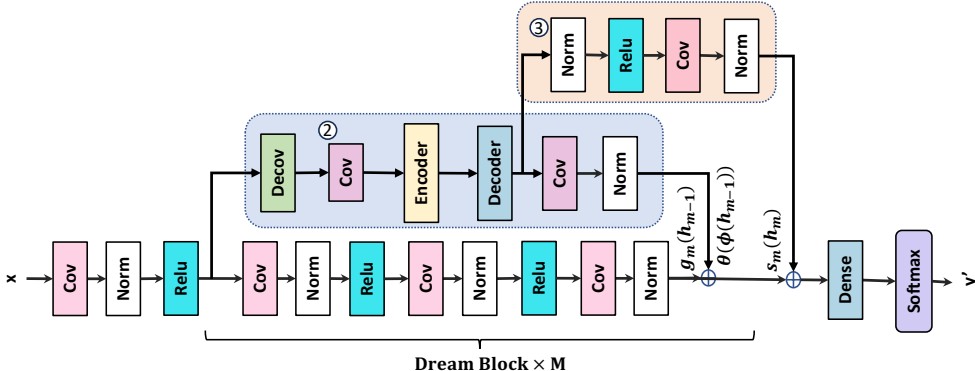

Figure 4: Overview of the Visual DreamNet Architecture, integrating "Dream Blocks" that include chain-like blocks processing data through convolutional layers ("Cov"), normalization ("Norm"), and the dream connection, enhanced with a full encoder-decoder setup for advanced feature consolidation and reconstruction, simulating "dream" states where the network reinterprets input data. The simulated "dreams" will be processed in block ③ with convolutional layers and pass to the dense layer for the final classification.

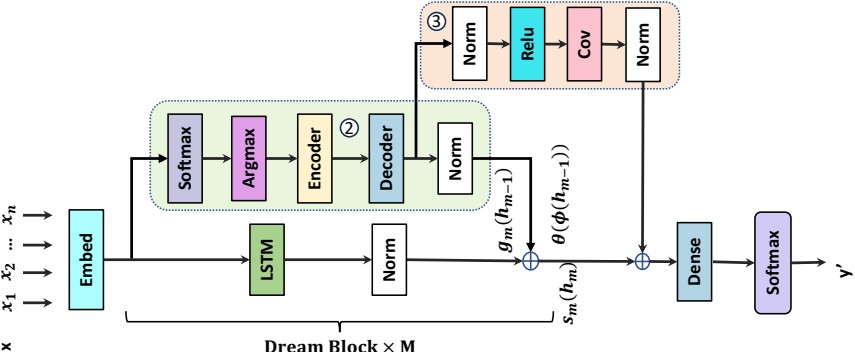

Figure 5: Overview of the Textual DreamNet Architecture, integrating "Dream Blocks" that include by chain-like blocks processing data through LSTMs, normalization ("Norm"), and the dream connection, enhanced with a full encoder-decoder setup for advanced feature consolidation and reconstruction, simulating 'dream' states where the network reinterprets input data. The simulated "dreams" will be processed in block ③ with convolutional layers and pass to the dense layer for the final classification.

blocks is then introduced into the dream connection (block ② in Fig 5), where it is first encoded into a latent space and subsequently decoded, enhancing the sequence with deeper insights and contextual understanding. This enriched sequence from the autoencoder is not only refined but also combined with outputs from other processing blocks in ③ in Fig 5. This integration stage involves additional LSTM processing to further enhance sequence quality and integrate the various feature enhancements comprehensively. The final output, a richly enhanced and consolidated feature set, then moves to a dense layer and a softmax classifier for final classification. DreamNet's capacity to iteratively enhance and integrate features in a simulated "dream" state leads to notable improvements in learning efficacy and classification accuracy across complex textual tasks. To give a more vivid illustration, we present Table 1 by showing the original texts and the generated dream-like text.

## 4 Experiments

We evaluate our models using public datasets and strong baselines. Our code and data are on GitHub[1].

---

[1] https://anonymous.4open.science/r/DreamNet-SleepNet-3B83

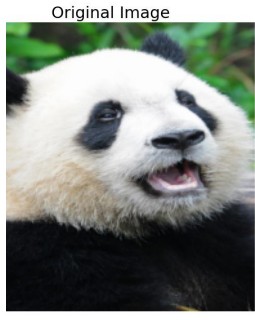 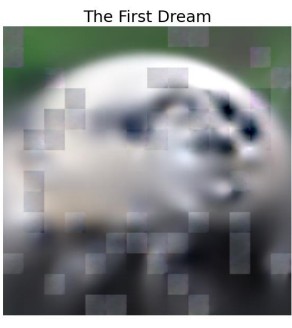 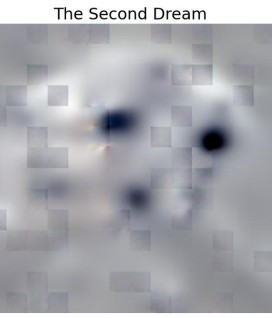 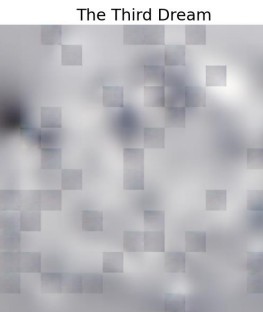

Figure 6: Stages of image transformation by DreamNet-3 using a masked autoencoder (MAE) He et al. (2021). Starting with the 'Original Image' of a panda National Geographic (2024), the sequence through "The First Dream", "The Second Dream", and "The Third Dream" illustrates progressive abstractions, depicting the model's process of deepening feature exploration and refinement.

Table 1: Dream Transformations: This table illustrates a sequence of textual transformations performed by DreamNet-3 on an excerpt about e-mail usage from the Ag News Business category. Each entry represents a stage in the text's progressive abstraction, demonstrating how the DreamNet-3 model reinterprets and abstracts the original content to explore deeper conceptual layers.

| Dream Transformation |
| --- |
| **Original Text**: Researchers seek to untangle the e-mail thread. E-mail is a victim of its own success. That's the conclusion of IBM Corp. researchers in Cambridge, who have spent nearly a decade conducting field tests at IBM and other companies about how employees work and use electronic mail. It's clear to them that e-mail has become the Internet's killer application. |
| **The First Dream**: From the silicon depths arises a storm of data, where electronic whispers coalesce into a thunderous echo of interactions, echoing in the vast void of virtual space. |
| **The Second Dream**: Echoes of former bytes, swirling chaos of the network's mind, a phantom web of disconnected truths and digital illusions |
| **The Third Dream**: Mirrors on keyboards, spinning yarns of byte-sized nonsense. E-melodies crooning to the rhythm of a chaotic data dance. |

## 4.1 Experimental Settings

To rigorously assess the efficacy of SleepNet and DreamNet, we test these models in both CV and NLP tasks. For CV tasks, we utilized datasets CIFAR100 Krizhevsky et al. (2009), ImageNet-tiny Le & Yang (2015), and ImageNet 1K Russakovsky et al. (2015). For NLP tasks, we employed datasets AG News Zhang et al. (2015a), IMDB Maas et al. (2011), and Yelp Zhang et al. (2015b). To facilitate a fair and detailed comparison, we varied the number of sleep/dream blocks ($M$) from 1 to 4. Specifically, SleepNet-1 denotes

Table 2: Overview of datasets used in the experiments

| Type | Dataset | Train Size | Test Size | #Classes |
| --- | --- | --- | --- | --- |
| Vision | CIFAR100 | 50,000 | 10,000 | 100 |
| | ImageNet-tiny | 100,000 | 1000 | 200 |
| | ImageNet-1k | 1,281,167 | 100,000 | 1000 |
| Language | AG News | 120,000 | 7600 | 4 |
| | IMDB | 25,000 | 25,000 | 2 |
| | Yelp | 650,000 | 5,000 | 5 |

Table 3: Performance comparison of SleepNet, DreamNet, and various baseline models on computer vision tasks, measured by accuracy. The table includes input size, parameter count, and FLOPs. The best results are highlighted in bold.

| Types | Models | Input Size | #Params | FLOPs | CIFAR100 | ImageNet tiny | ImageNet 1K |
|---|---|---|---|---|---|---|---|
| Convolution only | EfficientNet-B7 | $600^2$ | 66M | 37B | 90.1 | 80.1 | 84.7 |
| | EfficientNetV2-L | $480^2$ | 121M | 53B | 92.1 | 77.3 | 85.7 |
| | ResNet18 | $224^2$ | 11M | 1.8B | 80.6 | 68.9 | 69.7 |
| | ResNet50 | $224^2$ | 25M | 3.8B | 86.9 | 68.0 | 76.0 |
| | ResNet101 | $224^2$ | 45M | 7.6B | 84.1 | - | 80.8 |
| Transformer | $ViT_{base}$ | $224^2$ | 86M | 55.4B | 87.3 | 86.1 | 79.4 |
| | $ViT_{large}$ | $384^2$ | 307M | 190.7B | 91.0 | 88.0 | 83.0 |
| | $MAE_{base}$ | $224^2$ | 86M | 17.6B | 91.1 | 87.0 | 82.6 |
| | $MAE_{large}$ | $224^2$ | 304M | 61.9B | 91.3 | 87.1 | 83.1 |
| Convolution +Transformers | CvT-21 | $384^2$ | 32M | 24.9B | 90.1 | 83.1 | 83.3 |
| | CvT-W24 | $384^2$ | 277M | 193.2B | 92.1 | 88.1 | - |
| | CoAtNet-2 | $224^2$ | 75M | 15.7B | - | 87.1 | 84.1 |
| | CoAtNet-3 | $224^2$ | 168M | 34.7B | - | 87.6 | 84.5 |
| Augmentaion +Transformers | $ViT_l$-ACN | $384^2$ | 490M | 41.9B | 91.2 | - | 85.7 |
| | $MAE_l$-MAS | $224^2$ | 551M | 29.9B | 90.0 | - | 83.6 |
| SleepNet | SleepNet-2 | $224^2$ | 272M | 31.1B | 83.1 | 64.2 | 63.0 |
| | SleepNet-3 | $224^2$ | 272M | 39.4B | 92.2 | 88.1 | 85.9 |
| | SleepNet-4 | $224^2$ | 273M | 44.4B | 91.1 | 88.2 | 83.9 |
| | $SleepNet-3_{ViT-b}$ | $224^2$ | 272M | 39.4B | 92.2 | 88.1 | 85.9 |
| | $SleepNet-3_{ViT-l}$ | $224^2$ | 498M | 42.4B | 92.3 | 88.4 | 86.4 |
| | $SleepNet-3_{MAE-b}$ | $224^2$ | 272M | 40.1B | 90.2 | 86.1 | 81.9 |
| | $SleepNet-3_{MAE-l}$ | $224^2$ | 498M | 42.5B | 91.0 | 86.3 | 84.1 |
| DreamNet | DreamNet-2 | $224^2$ | 732M | 30.0B | 85.1 | 71.9 | 71.6 |
| | DreamNet-3 | $224^2$ | 733M | 42.1B | 92.3 | 89.1 | 87.8 |
| | DreamNet-4 | $224^2$ | 733M | 50.5B | 85.1 | 89.9 | **89.2** |
| | $DreamNet-3_{MAE-b}$ | $224^2$ | 733M | 42.1B | 92.3 | 89.1 | 87.8 |
| | $DreamNet-3_{MAE-l}$ | $224^2$ | 910M | 51.1B | **93.4** | **89.6** | 88.9 |

a network configuration with one sleep block, SleepNet-2 with two blocks, and so forth, up to SleepNet-4. Similarly, DreamNet-1 to DreamNet-4 follow the same naming convention.

To ensure an equitable comparison, identical training protocols were maintained across all models. Each classifier underwent training for 30 epochs using the ADAM optimizer Kingma & Ba (2014). The training parameters were set as follows: a learning rate ($lr$) of 0.005, with the coefficients for running averages of the gradient and its square set at $\beta_1 = 0.9$ and $\beta_2 = 0.999$, respectively. Additionally, a denominator ($\sigma$) of $10^{-5}$ was introduced to bolster numerical stability. This consistent training approach allows for a rigorous evaluation of each model's performance, ensuring that any observed differences are attributable to the model architectures rather than discrepancies in training procedures.

Table 4: Performance comparison of SleepNet, DreamNet, and various baseline models on linguistic tasks, measured by accuracy. The table includes token size, parameter count, and FLOPs. The best results are highlighted in bold.

| Types | Models | Vocab Size | #Params | FLOPs | AG News | IMDB | Yelp |
|---|---|---|---|---|---|---|---|
| Convolution LSTM | WordCNN | 800,000 | 1.76M | 0.02B | 81.7 | 72.8 | 87.8 |
| | LSTM-2 | 800,000 | 1.3M | 0.01B | 80.0 | 42.2 | 77.1 |
| | LSTM-3 | 800,000 | 1.4M | 0.01B | 82.0 | 52.2 | 81.1 |
| | LSTM-4 | 800,000 | 1.5M | 0.01B | 84.5 | 71.1 | 90.3 |
| Transformer | RoBERTa$_{base}$ | 50,265 | 125M | 11.2B | 89.1 | 91.1 | 94.7 |
| | RoBERTa$_{large}$ | 50,265 | 355M | 39.5B | 92.3 | 85.3 | 95.9 |
| | DistillBERT$_{base}$ | 30,522 | 66M | 5.6B | 85.3 | 84.7 | 90.3 |
| | XLNet$_{base}$ | 32,000 | 117M | 13.3B | 90.1 | 90.1 | 93.1 |
| | XLNet$_{large}$ | 32,000 | 360M | 360.3B | 92.0 | 93.2 | 97.1 |
| SleepNet | SleepNet-1 | 50,265 | 210M | 27.4B | 83.4 | 60.9 | 70.9 |
| | SleepNet-2 | 50,265 | 211M | 21.3B | 88.3 | 73.3 | 90.2 |
| | SleepNet-3 | 50,265 | 211M | 25.3B | 92.3 | 90.1 | 93.2 |
| | SleepNet-4 | 50,265 | 213M | 30.1B | 93.3 | 92.8 | 94.3 |
| | SleepNet-3$_{RoBERTa-b}$ | 50,265 | 351M | 27.1B | 91.3 | 93.3 | 96.9 |
| | SleepNet-3$_{RoBERTa-l}$ | 50,265 | 589M | 41.2B | 91.4 | 94.1 | 97.5 |
| | SleepNet-3$_{XLNet-b}$ | 32,000 | 331M | 43.7B | 92.0 | 90.1 | 93.4 |
| | SleepNet-3$_{XLNet-l}$ | 32,000 | 371M | 49.1B | 92.1 | 92.1 | 95.7 |
| DreamNet | DreamNet-1 | 50,265 | 435M | 27.4B | 88.0 | 77.9 | 81.9 |
| | DreamNet-2 | 50,265 | 436M | 27.4B | 92.5 | 83.0 | 90.8 |
| | DreamNet-3 | 50,265 | 436M | 27.4B | 93.3 | 91.1 | 95.0 |
| | DreamNet-4 | 50,265 | 437M | 27.4B | **94.4** | 93.4 | 95.5 |
| | DreamNet-3$_{RoBERTa-b}$ | 50,265 | 471M | 27.4B | 91.3 | 90.1 | 95.1 |
| | DreamNet-3$_{RoBERTa-l}$ | 50,265 | 752M | 27.4B | 93.3 | 91.0 | **97.9** |
| | DreamNet-3$_{XLNet-b}$ | 32,000 | 451M | 27.4B | 93.5 | 91.0 | 92.9 |
| | DreamNet-3$_{XLNet-l}$ | 32,000 | 781M | 27.4B | 95.3 | **95.1** | 95.9 |

## 4.2 Augmentation Methods

Data augmentation techniques are crucial for enhancing model performance in computer vision tasks and are widely applied to state-of-the-art visual models. To ensure robust comparisons, we incorporated two powerful data augmentation methods, namely RandAugment Cubuk et al. (2020) and Mixup Zhang et al. (2017), into our visual models and the baselines. RandAugment systematically searches for the best augmentation policies by randomly applying a fixed number of distortions, optimizing the augmentation strategy for better generalization. Mixup, on the other hand, creates new training examples by combining pairs of images and their labels, encouraging models to behave linearly in between training examples and improving robustness against adversarial examples.

In natural language processing (NLP), language models are typically trained with augmentation techniques from methods such as the TextAttack Morris et al. (2020) framework, since the original datasets may not be sufficiently large to achieve reasonable performance on their own Chen et al. (2023). Specifically, TextAttack augmentation employs a combination of word swaps, insertions, and substitutions to generate new examples. In our setting, we augmented the text data (for all models in comparison) by editing 10% of the words in each instance per augmentation, effectively doubling the size of the training set with these new variations. This approach ensures that the models are exposed to a variety of linguistic variations, enhancing their ability to generalize and perform well on diverse text inputs.

### 4.3 Datasets and Metrics

In this subsection, we describe the datasets and metrics used in our experiments for visual and language tasks, and specify the performance evaluation metrics.

**Datasets**

To ensure thorough evaluation, we used diverse datasets for visual tasks, including CIFAR-100 Krizhevsky et al. (2009), ImageNet-tiny Le & Yang (2015), and ImageNet-1K Russakovsky et al. (2015), and for language tasks, including Ag News Zhang et al. (2015a), IMDB Maas et al. (2011), and Yelp Zhang et al. (2015b). This selection ensures a comprehensive assessment, highlighting the versatility and effectiveness of SleepNet and DreamNet in both visual and language tasks. Dataset specifics are in Table 2.

**Metrics**

To evaluate model performance, we use classification accuracy, model size, and complexity, measured by input size, number of parameters, and FLOPs. The metrics are:

- **Accuracy**: Correct predictions divided by total predictions.
- **Input Size** (CV models): Dimensions of images fed into the model, affecting detail and computational resources.
- **Vocabulary Size** (NLP models): Number of unique tokens, impacting language representation and computational demands.
- **Number of Parameters (#Params)**: Total trainable weights, reflecting model complexity and learning capacity.
- **FLOPs**: Floating-point operations per prediction, indicating computational complexity.

### 4.4 Baselines

During the evaluation of our models, we benchmarked their performance against various state-of-the-art models to ensure comprehensive comparisons. For vision-focused tasks, we utilized:

- **ResNet variants** He et al. (2015): ResNet18 and ResNet50, known for their deep residual learning capabilities in image classification.
- **EfficientNet** Tan & Le (2019): Models that balance network depth, width, and resolution for optimal performance using compound scaling.
- **Attention-based Transformers**: Vision Transformer (ViT) Dosovitskiy et al. (2020a) and ViT-G Zhai et al. (2022), leveraging self-attention for processing image patches.
- **Hybrid Methods**: CoAtNet Dai et al. (2021) and CvT Wu et al. (2021), combining CNNs with attention mechanisms for efficient image processing.
- **Augmentation-based Models**: Augmenting Convolutional Network (ACN) Touvron et al. (2021) and Masked Augmentation Model (MAS) Heo et al. (2023), enhancing feature selection during training.

For text tasks, we benchmarked our models against:

- **Vanilla LSTM**: A basic Long Short-Term Memory network for sequential data and text processing.
- **TextCNN** Kim (2014): A convolutional neural network for text classification, capturing different n-grams.

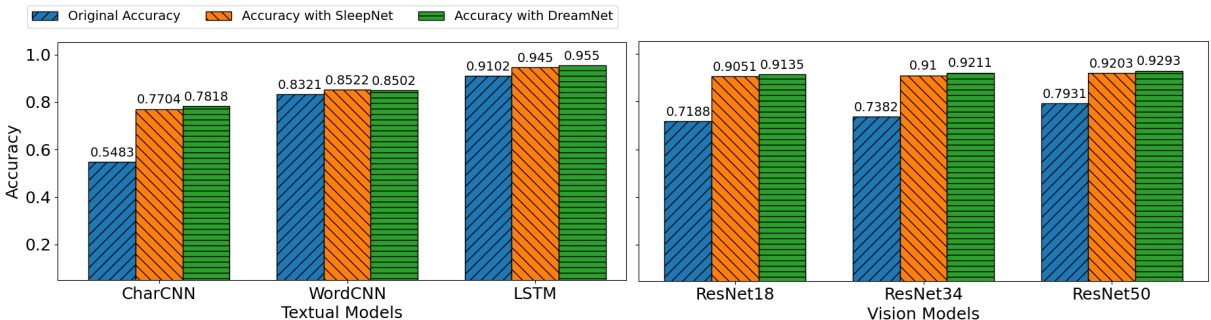

Figure 7: Ablation studies for testing the different chain-like blocks by comparing the original performance of various textual and vision classifiers against their performance when integrated with the proposed methods, SleepNet and DreamNet.

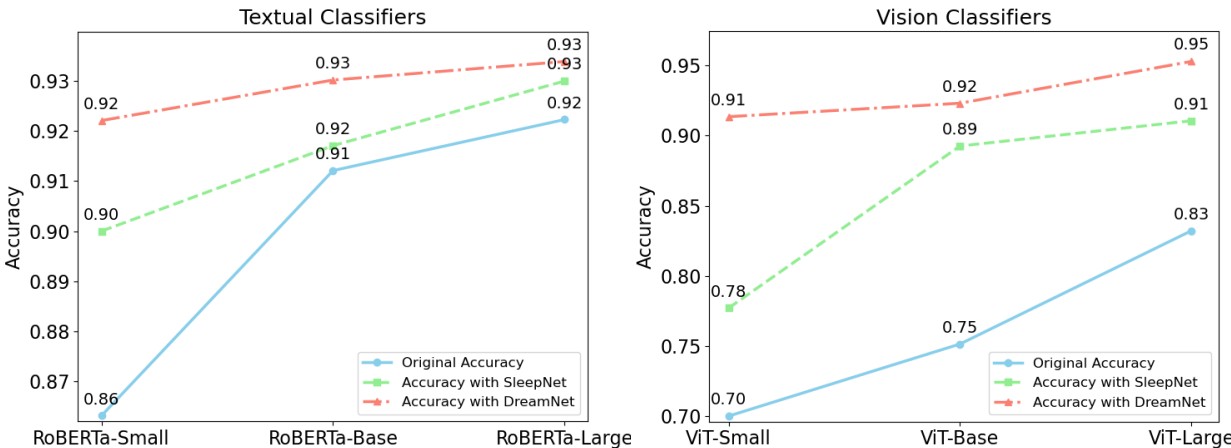

Figure 8: Ablation studies for testing the different pre-trained encoders/autoencoders by comparing the original performance of various textual and vision classifiers against their performance when integrated with the proposed methods, SleepNet and DreamNet.

- **Transformer-based Models**: XLNet Yang et al. (2019) for bidirectional context, RoBERTa Liu et al. (2019) with optimized training, and DistilBERT Sanh et al. (2019), a smaller, efficient version of BERT.

### 4.5 Main Results and Analysis

The main results of the experiments are presented in Tables 3 and 4. In the evaluation of computer vision tasks, vision transformers, particularly MAE-large, demonstrated strong performance, achieving 91.3% on CIFAR100 and 87.1% on ImageNet-tiny. Hybrid models, such as CvT-W24 and CoAtNet-3, also performed impressively across all datasets, highlighting the benefits of integrating convolutional and attention mechanisms. Notably, DreamNet-3$_{\text{MAE-l}}$ achieved the highest accuracy on CIFAR100 (93.4%) and Tiny-ImageNet (89.6%), while DreamNet-4 recorded the best results on ImageNet 1K (89.2%). Additionally, SleepNet consistently outperformed the baselines, securing the second-best results and displaying a performance closely competitive with that of DreamNet.

The superior performance of DreamNet over SleepNet and other models can be attributed to several factors. The dream connection in DreamNet refines and consolidates features learned during training, enhancing the model's ability to generalize to new data. This mechanism allows DreamNet-3 to achieve the highest accuracy on ImageNet 1K (87.8%) and CIFAR100 (92.3%). Additionally, the use of pre-trained self-supervised

Table 5: Efficiency is evaluated by the hour per epoch on CIFAR100 and ImageNet. The best-performed model is highlighted in bold. EffNetB7 refers to EfficientNet-B7.

| Datasets | ResNet18 | ResNet50 | EffNetB7 | $ViT_{base}$ | $MAE_{base}$ | CoAtNet2 | CvT-21 | SleepNet | DreamNet |
|---|---|---|---|---|---|---|---|---|---|
| CIFAR100 | **0.45** | 0.65 | 0.55 | 1.04 | 0.73 | 1.14 | 1.23 | 0.8 | 1.8 |
| ImageNet tiny | **6.77** | 7.43 | 7.11 | 10.60 | 8.60 | 14.60 | 13.60 | 8.81 | 18.9 |

encoders in both SleepNet and DreamNet provides a strong initialization, further improving their feature extraction capabilities. SleepNet-3$_{MAE-l}$ set a new benchmark on CIFAR100 with an accuracy of 93.4%, highlighting the significant improvements brought by the "dream" mechanism in enhancing learning and generalization.

For natural language processing tasks, both SleepNet and DreamNet surpassed these baselines. Specifically, DreamNet achieved the highest performance across all datasets, recording 94.4% on AG News with DreamNet-4, 95.1% on IMDB with SleepNet-3$_{XLNet-l}$, and 97.9% on Yelp with SleepNet-3$_{RoBERTa-l}$. Additionally, SleepNet consistently outperformed other models, securing the second-best results and displaying a performance that was closely competitive with that of DreamNet.

## 4.6 Ablation Studies

Since DreamNet is built on top of SleepNet, in our ablation studies, we begin by comparing SleepNet and DreamNet models and analyzing their individual advantages. Next, we shall explore alternative chain-like blocks and pre-trained encoders/autoencoders. Then, we shall test whether the parameters of the pre-trained models should be frozen.

### 4.6.1 Performance comparison between DreamNet and SleepNet.

In both CV and NLP tasks, our experiments were documented with results presented in Tables 3 and 4. The findings demonstrate that DreamNet consistently outperformed SleepNet across various datasets. These results highlight DreamNet's consistent edge in both visual and textual data tasks.

The superior performance of DreamNet can be attributed to several key factors. Firstly, the "dream" blocks in DreamNet play a crucial role in refining and consolidating features learned during training. This iterative refinement allows DreamNet to capture more intricate patterns and improve its generalization capabilities. Secondly, DreamNet leverages pre-trained self-supervised encoders, which provide a robust initialization and enhance feature extraction. This strong foundation significantly boosts the model's ability to learn from complex datasets. Additionally, the integration of "dream" blocks enables DreamNet to iteratively refine features, leading to superior performance on both CV and NLP tasks. The advanced "dream" connection mechanism, as seen in models like DreamNet-3$_{RoBERTa-l}$, further enhances feature learning and generalization, enabling DreamNet to consistently outperform SleepNet and other baseline models.

### 4.6.2 Comparisons with different numbers of sleep/dream blocks

We investigate the impact of varying the number of sleep and dream blocks on the performance of SleepNet and DreamNet. As depicted in Tables 3 and 4, there is a noticeable trend in the performance improvement with the increase in the number of sleep and dream blocks. For CV tasks, models with more blocks generally achieved higher accuracy. Specifically, on CIFAR100, SleepNet-4 and DreamNet-4 showed significant gains, with DreamNet-4 reaching up to 92.3% accuracy. Similarly, on ImageNet-tiny, DreamNet-4 outperformed the other configurations, achieving the highest accuracy. A similar pattern was observed in NLP tasks. On the AG News dataset, DreamNet-4 achieved the highest accuracy of 94.4%, demonstrating the effectiveness of additional blocks. Both SleepNet and DreamNet benefited from increasing the number of blocks, although DreamNet consistently maintained a performance edge.

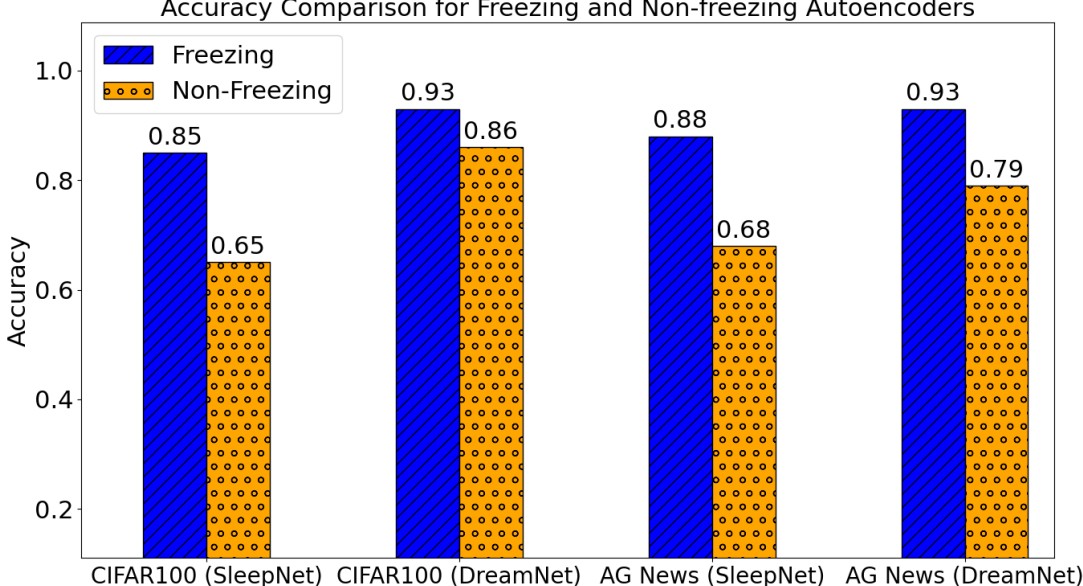

Figure 9: This plot delineates the impact of freezing (marked in blue) and unfreezing (indicated in orange) the parameters of unsupervised models on AG News and CIFAR100.

The observed performance improvements with increasing numbers of sleep and dream blocks can be attributed to several factors. Firstly, adding more blocks allows the model to capture more complex features and hierarchical patterns in the data. This is particularly beneficial for tasks with high variability and intricate structures, such as those found in CV and NLP datasets. Secondly, the pre-trained self-supervised encoders used in both SleepNet and DreamNet provide a strong initial representation, which is further improved through the added blocks. This architectural design enables the models to leverage deep learning's hierarchical nature effectively, resulting in superior performance as the depth increases. The consistent outperformance of DreamNet over SleepNet highlights the significant contribution of the dream mechanism in refining and consolidating features, making it a crucial component in achieving high accuracy across various tasks.

### 4.6.3 Comparisons with different chain-like models.

We evaluated the efficacy of various supervised models using three distinct text-based chain-like blocks: two-layer LSTMs, CharCNN, and WordCNN. CharCNN and WordCNN, variants of TextCNN, differ primarily in their tokenization units. For the vision tasks, we selected ResNet18, ResNet34, and ResNet50, which vary in the number of convolutional layers. Fig 7 summarizes their performance on the AG News dataset for text tasks and CIFAR100 for vision tasks. There is a clear trend that shows that the better the performance of the chain-like block, the better the subsequent performance of the proposed models (SleepNet and DreamNet). This correlation suggests that stronger initial performance from a classifier enhances its subsequent integration into SleepNet, likely due to the foundational role of chain-like blocks in the models.

### 4.6.4 Comparisons with different unsupervised encoders and autoeconders.

We assessed the performance of SleepNet and DreamNet by utilizing various unsupervised encoders and autoencoders, as depicted in Fig 8. Incorporating more sophisticated encoders consistently improved results: employing RoBERTa-Small with SleepNet increased accuracy from 0.86 to 0.90, and with DreamNet, it reached 0.92. Similarly, ViT-Small improved from 0.70 to 0.78 with SleepNet and to 0.91 with DreamNet. This trend demonstrates that the more advanced the encoder, the better the performance of the proposed methods. The primary reason for this improvement is that sophisticated encoders provide richer feature

representations, enhancing the learning process. SleepNet benefits from the additional feature extraction capabilities, while DreamNet's dream blocks further refine and consolidate these features, leading to superior performance.

### 4.6.5 Comparisons with frozen and unfrozen unsupervised encoders/autoencoders.

Equally importantly, we examined the effect of freezing versus unfreezing the parameters of unsupervised encoders/autoencoders. This evaluation was carried out across three distinct tasks, employing varied models: TextCNN and BERT-based SleepNet for text classification on the AG News, and ResNet18 coupled with Google's ViT Base for image classification on CIFAR100. Fig 9 demonstrates that consistently across these varied tasks and models, the "Frozing" configuration (depicted in blue) outperforms the "Non-Freezing" one (shown in orange). For computer vision tasks, models with frozen encoders generally achieved higher accuracy. For instance, DreamNet with unfrozen MAE encoders reached 93% accuracy on CIFAR100, compared to 86% with frozen encoders. In the NLP domain, a similar trend was observed. On the AG News dataset, SleepNet-3 with unfrozen BERT encoders achieved an accuracy of 88%, outperforming the unfrozen encoder version, which attained 68%. These results demonstrate that frozen encoders generally provide better performance across both CV and NLP tasks.

We attribute this consistently superior performance of frozen encoders to two major reasons. Firstly, unfreezing the parameters during training may often tend to overfit the model. The overfitting can be traced back to finding an optimal learning rate for such a setup. More specifically, the supervised component of the model, which is not pre-trained, requires a larger learning rate to capture complex patterns effectively. In contrast, the pre-trained unsupervised part necessitates a lower learning rate to avoid drastic changes that can degrade the valuable pre-trained patterns. Balancing this diverse learning rate needs while unfreezing parameters is non-trivial and often leads to overfitting. Secondly, since the unsupervised models contribute additional features to the supervised models, any parameter alteration could modify these supplementary features to fit the specific dataset being processed. Although this might seem beneficial, it could inadvertently filter out some of the generalized, useful latent information that the unsupervised encoder initially captured, thereby limiting the model's overall ability to generalize across diverse datasets.

### 4.7 Complexity and Qualitative Results

We conducted our experiments on a RHEL 7.9 system equipped with an Intel(R) Xeon(R) Gold 6238R CPU, an NVIDIA Quadro RTX 5000 GPU, and 88GB RAM. Table 5 presents a comparison of the computational efficiency of different models, evaluated in terms of hours per epoch on the CIFAR100 and ImageNet-tiny datasets. ResNet18 was the most efficient, requiring only 0.45 hours per epoch on CIFAR100 and 6.77 hours per epoch on ImageNet-tiny. SleepNet also demonstrated good efficiency, surpassing ViT-B with 0.8 hours per epoch on CIFAR100 and 8.81 hours per epoch on ImageNet-tiny, compared to ViT-B's 1.04 and 10.60 hours, respectively. This shows that SleepNet is computationally optimized despite its advanced features.

Analyzing the FLOPs from Tables 3 and 4, ResNet18 had the lowest complexity with 1.8 billion FLOPs. In contrast, DreamNet-4 had 50.5 billion FLOPs, achieving higher accuracy (92.3% on CIFAR100 and 89.9% on ImageNet-tiny). For NLP tasks, DreamNet-4, with 42.1 billion FLOPs, achieved 94.4% accuracy on AG News and 92.3% on IMDB, outperforming models with fewer FLOPs. This illustrates the trade-off between computational complexity and performance, highlighting DreamNet's ability to leverage its higher complexity for superior accuracy.

Overall, SleepNet and DreamNet effectively balance computational efficiency and performance. DreamNet, although more complex, provides notable accuracy improvements, making it suitable for tasks where performance is critical. The integration of sleep and dream blocks in both models demonstrates that added complexity translates into significant performance gains across computer vision and natural language processing tasks.

## 5 Conclusion and Future Work

In this paper, we introduced SleepNet and DreamNet, two innovative deep learning architectures inspired by cognitive processes in biological brains. These models use novel sleep and dream mechanisms to consolidate and refine features, leading to improved performance in both computer vision and natural language processing tasks. Our experiments show that these models outperform state-of-the-art results, highlighting their effectiveness and potential for general applicability.

Our future work includes optimizing SleepNet and DreamNet for better efficiency and performance, integrating additional cognitive-inspired mechanisms, and applying these models to a broader range of tasks and datasets. We also plan to explore their potential in real-time applications and scalability to more complex tasks, aiming to develop more intelligent and efficient AI systems.

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
