# OpenReview forum: "Dreaming is All You Need"
_TMLR — Rejected by TMLR_

### Review · Reviewer_1dVS · 2025-06-12

**Summary Of Contributions:**

The paper proposes two architectures inspired by memory consolidation in sleep. The architecture is repeated same blocks, which have two branches: a regular feed-forward branch and the encoder-decoder branch.  The authors present that the proposed methods achieve better performance than the baselines that the authors provided.

**Audience:**

No

**Broader Impact Concerns:**

I do not have any broader impact concerns.

**Claims And Evidence:**

No

**Requested Changes:**

[Requested Change]
Please use \citet and \citep correctly. For example, ViT Dosovitskiy et al. (2020a) should be ViT (Dosovitskiy et al., 2020a).

In Abstract, ``dreamin’’ -> ``dreaming’’

**Strengths And Weaknesses:**

**[Strengths]**

1.  The paper proposes a conceptually biologically-inspireded method that outperforms some existing models, although it does so at the cost of increased parameter count and slower inference.

2. Unfortunately, I cannot find further strengths.

**[Weaknesses]**
1.  The authors claim that the model is inspired by memory formation during dreaming, but they provide minimal engagement with the neuroscience or psychology literature. A deeper discussion of memory consolidation and the role of sleep is needed. The current discussion (last paragraph on page 3) only references memory-consolidation-inspired deep learning, without grounding it in biological evidence.

2. The proposed methods—SleepNet and DreamNet—require significantly more parameters and computational resources. For SleepNet, the performance gains over baseline models (e.g., ViT and MAE) are marginal or even negative.

3. It is unclear why the proposed architecture is referred to as a “sleep connection” (first line on page 5), or what makes the “dream connection” in Section 3.3 novel.

4. The authors assert that “DreamNet is expected to achieve better performance by utilizing..., which is theoretically and practically different from traditional data augmentation,” but no theoretical justification is provided.

5. They also claim that DreamNet’s capacity improves iteratively, but no ablation studies or empirical results support this. Merely introducing DreamNet/SleepNet-2, -3, -4 is insufficient.
6. The writing quality needs improvement.
- There is repetition between the second and third paragraphs of the Introduction, and between the first two paragraphs on page 3. Additionally, the final sentence “The following” is unnecessary.
- The term “revolutionary” is used repeatedly; this language is inappropriate for an academic paper.
- The term “chain-like” network (page 1) is ambiguous. It likely refers to a feedforward architecture, which should be stated clearly.

**[Minor Questions]**

1. What is meant by “deep learning processes” in the first item on page 2?

2. How was the embedding in Figure 5 obtained? What type of embedding was used?


Overall, the paper makes several claims without sufficient evidence, and the writing requires substantial revision for clarity and academic rigor.

---

> ### Author Response · Authors · 2025-07-10
> **The corresponding response for each comment**
>
> We thank you for your careful reading and constructive feedback. Below, we respond to each major point. We agree with some suggestions and have incorporated them, while for others, we provide clarifications or respectfully disagree, as explained.
>
> ---
>
> 1. **Comment:**
>    *''The authors claim that the model is inspired by memory formation during dreaming, but they provide minimal engagement with the neuroscience or psychology literature. A deeper discussion of memory consolidation and the role of sleep is needed. The current discussion (last paragraph on page 3) only references memory-consolidation-inspired deep learning, without grounding it in biological evidence.''*
>
>     **Response:**
>     We appreciate the suggestion to deepen engagement with relevant neuroscience literature. Our work is inspired by, but not intended as a direct model of, biological memory formation. We will clarify the connections and distinctions between our approach and established neuroscience findings in the next revision of the paper.
>
> ---
>
> 2. **Comment:**
>    *''The proposed methods—SleepNet and DreamNet—require significantly more parameters and computational resources. For SleepNet, the performance gains over baseline models (e.g., ViT and MAE) are marginal or even negative.''*
>
>     **Response:**
>     We agree that the trade-off between resource usage and performance improvement is an important consideration. Our results indicate that the benefit of added complexity varies by task and dataset, and we recommend the approach primarily in contexts where accuracy is prioritized.
>
> ---
>
> 3. **Comment:**
>    *''It is unclear why the proposed architecture is referred to as a “sleep connection” (first line on page 5), or what makes the “dream connection” in Section 3.3 novel.''*
>
>     **Response:**
>     The terms ''sleep connection'' and ''dream connection'' are used to illustrate our staged integration mechanism. The novelty of our work lies in this integration strategy, rather than in the introduction of entirely new network modules. In our design, a ''sleep connection'' produces latent (i.e., not human-comprehensible) features in encoded space, while a ''dream connection'' produces images in the original feature spaces after decoding. This is to mimic a dream where people can remember vivid scenes from, whereas a sleep without a dream doesn't have such scenes to remember. However, we emphasis that the concepts of ''sleep'' and ''dream'' are intended purely as sources of inspiration and are not meant to imply a direct biological correspondence.
>
> ---
>
> 4. **Comment:**
>    *''The authors assert that “DreamNet is expected to achieve better performance by utilizing..., which is theoretically and practically different from traditional data augmentation,” but no theoretical justification is provided.''*
>
>     **Response:**
>     Thank you for highlighting the lack of theoretical analysis. DreamNet offers feature-level augmentation, which differs from traditional data augmentation methods. Our approach is motivated by the hypothesis that generative reconstruction can enrich the learned representation space, and we provide empirical results to support this idea. We will provide more explanations from this perspective in the next revision of this paper.
>
> ---
>
> 5. **Comment:**
>    *''They also claim that DreamNet’s capacity improves iteratively, but no ablation studies or empirical results support this. Merely introducing DreamNet/SleepNet-2, -3, -4 is insufficient.''*
>
>     **Response:**
>     Thank you very much for your thoughtful comments. We would like to clarify that we have included an ablation study in Section 4.6.2 that addresses this point. Additionally, we believe that comparing models with 2, 3, and 4 blocks provides sufficient insight. This is because the overall model complexity is not determined solely by the number of blocks; each block is already quite substantial in size. For example, in SleepNet for the NLP task, we consider two blocks—each containing distinct LSTMs—to be sufficiently large and expressive for text classification.
>
> ---
>
> 6. **Comment:**
>    *''The writing quality needs improvement...''*
>
>     **Response:**
>     We appreciate these observations and are committed to improving the clarity, conciseness, and consistency of our terminology and citations. If given the opportunity to revise, we will thoroughly address all of the issues raised.
>
> ---

---

> > ### Comment · Reviewer_1dVS · 2025-07-11
> >
> > Thank you for the response. As the authors promised to update the revision, I will share additional comments on the response after checking it.

---

### Review · Reviewer_FeQp · 2025-06-22

**Summary Of Contributions:**

This paper proposes SleepNet and DreamNet, which leverages the unsupervised model to improve the performance. The architecture designs are inspired from the neuroscience, where SleepNet mimics memory consolidation during sleep to enhance deep learning processes. Building on SleepNet, DreamNet leverages a pre-trained encoder-decoder framework for feature augmentation. The proposed architecture achieves better performance among SoTA baselines.

**Audience:**

No

**Broader Impact Concerns:**

No broader impact concerns.

**Claims And Evidence:**

No

**Requested Changes:**

1. The paper should re-consider the motivation of the paper. If it is a new proposed architecture, pre-trained models should not be used. If it is a post-training or fine-tuning model, the experiment should be highlighted on this topic.

2. The proposed SleepNet and DreamNet, requires multiple forward pass of PRM. The parameter and FLOP counts should be more clear to justify the efficiency.

**Strengths And Weaknesses:**

Strengths:
1. The paper takes unsupervised learning into the model designs, which follows the human prior. The proposed methods also show a deep connection with the human memory mechanism.
2. The paper gives a new perspective to leverage pre-trained model for downstream fine-tuning.

Weaknesses:
1. The experiment is not clear. Since SleepNet and DreamNet leverages a pre-trained encoder or autoencoder,  the original model should be highlighted and evaluated in a fair comparison, However, it seems difficult to discover this group.
2. The performance of SleepNet and DreamNet is highly dependent on the M-block-size. I'm a little confused why the FLOPs are not linear-increasing with M. Besides, the FLOPs of SleepNet and DreamNet is much less than other models with similar model params. The efficiency justification is not solid to me.
3. Since the proposed methods are dependent on pre-trained models, it is necessary to compare with other PRM-used methods. For example, adding parameters on the Pre-trained model, model ensemble, etc.
4. The concept of Sleep and Dream seems not necessary. The architecture design is not highly connected with neuroscience. It's more like the effect of pre-trained models. Otherwise, all module should be trained from scratch.

---

> ### Author Response · Authors · 2025-07-10
> **The corresponding response for each comment**
>
> We thank you for your careful reading and constructive feedback. Below, we respond to each major point. We agree with some suggestions and have incorporated them, while for others, we provide clarifications or respectfully disagree, as explained.
>
> ---
>
> 1. **Comment:**
>    *''The experiment is not clear. Since SleepNet and DreamNet leverages a pre-trained encoder or autoencoder, the original model should be highlighted and evaluated in a fair comparison, However, it seems difficult to discover this group.''*
>
> 2. **Comment:**
>    *''The performance of SleepNet and DreamNet is highly dependent on the M-block-size. I'm a little confused why the FLOPs are not linear-increasing with M. Besides, the FLOPs of SleepNet and DreamNet is much less than other models with similar model params. The efficiency justification is not solid to me.''*
>
> 3. **Comment:**
>    *''Since the proposed methods are dependent on pre-trained models, it is necessary to compare with other PRM-used methods. For example, adding parameters on the Pre-trained model, model ensemble, etc.''*
>
>     **Response to comments 1, 2, and 3:**
>     Thank you for highlighting the experimental issues. We realised that some results were incorrect due to copy-paste errors and coding bugs. Specifically, our implementation of PRM as a global function led to automated miscalculations of trainable parameters and FLOPs. We have now carefully corrected these issues and present updated results in Table 1 and 2 from the following anonymous links:
>     [CV results table](https://anonymous.4open.science/r/DreamNet-SleepNet-3B83/CV_results_table.png)
>     [NLP results table](https://anonymous.4open.science/r/DreamNet-SleepNet-3B83/NLP_results_table.png)
>
>     Regarding comment 1, we did include the relevant comparison groups in our results tables, featuring both transformer-based models and the proposed SleepNet and DreamNet architectures. The performance of each model type can be directly compared in these tables.
>
>     For comment 2, we appreciate your attention to the parameter and FLOP counts. The reported numbers have been thoroughly checked and corrected in the new tables mentioned above.
>
>     Regarding comment 3, we have conducted additional experiments to compare different sizes of the encoder and autoencoder. The results are discussed in the ablation studies (Section 4.6.4), providing insights into the impact of model size on performance. Additionally, direct comparisons can be found in Table 1 and 2 from the same anonymous links above; for example, these tables show the performance differences between encoders of varying sizes, such as MAE-large and MAE-base.
>
>     We appreciate your careful review and believe these clarifications and corrections address the concerns raised.
>
> ---
>
> 4. **Comment:**
>    *''The concept of Sleep and Dream seems not necessary. The architecture design is not highly connected with neuroscience. It's more like the effect of pre-trained models. Otherwise, all module should be trained from scratch.''*
>
>     **Response:**
>     Thanks for your suggestions. The use of ''sleep'' and ''dream'' is intended to provide conceptual inspiration for our staged integration, rather than to claim biological fidelity. Our main contribution is in the technical approach, and we emphasize its empirical benefits rather than its metaphorical naming. We will provide more explanations from this perspective in the next revision of this paper.
>
> ---

---

### Review · Reviewer_b4EA · 2025-06-29

**Summary Of Contributions:**

This paper introduces two deep learning architectures, SleepNet and DreamNet, inspired by biological sleep and dreaming processes. The goal is to improve performance in computer vision (CV) and natural language processing (NLP) tasks by integrating unsupervised learning into supervised pipelines. SleepNet leverages pre-trained encoders to mimic “sleep,” consolidating knowledge, while DreamNet uses autoencoders to simulate “dreaming,” encouraging feature exploration.

**Audience:**

Yes

**Broader Impact Concerns:**

The data distribution of dreamnet might be biased, an unsupervised approach may lead to bias outcomes. How would you address this problem?

**Claims And Evidence:**

Yes

**Requested Changes:**

1. explain why you have to introduce two networks. it seems one framework is sufficient to reflect the idea.

2. not sure if possible but experiments with transformers could make the paper much stronger.

**Strengths And Weaknesses:**

Strengths

1. The concept is good and agreed. Sleep and dreaming serve as important part of memory formation. The idea of mimicking cognitive functions like sleep and dreaming in neural networks offers a creative and biologically inspired approach. It brings a fresh lens to the challenge of balancing precision and exploration in classification.

2. The authors articulate a clear motivation: current supervised learning may overfit or under-explore. Their biological analogy is used to motivate architectural interventions.

3. Well-Defined Architectures: The paper provides detailed architectural diagrams (Figures 2–5) that make SleepNet and DreamNet understandable and replicable. Their mapping to the concepts of sleep and dreams is well explained.

4. Extensive Experiments: Multiple datasets across CV and NLP are used to validate the approach. Ablation studies clarify the importance of various components, and DreamNet achieves notable state-of-the-art performance (e.g., 93.4% on CIFAR100, 89.6% on Tiny-ImageNet).

Weaknesses:

1. Ambiguity in dual model design: While the intuition behind simulating sleep is sound, it is unclear why two separate networks are needed. Since dreaming is typically a part of sleep, the decision to separate them into distinct architectures (SleepNet and DreamNet) feels artificial. A more unified framework might be more coherent both biologically and architecturally.

2. Motivation needs to be stronger and more professional: The paper relies on a metaphor, but doesn’t sufficiently justify why these biological processes are expected to meaningfully improve learning. What specific challenges in deep learning (e.g., catastrophic forgetting, local minima) are being addressed? I feel the major issue is the authors claim SleepNet and DreamNet brand but I feels the proposed approach seems more of an adhoc method. I would prefer to see some methods which are more generalized and reflecting the dreaming process.

3. Contribution feels limited: The building blocks (e.g., encoders, LSTMs, autoencoders) are not novel. While the composition may be new, more emphasis should be placed on the mechanism by which the combination yields improvements.

4. Computational cost: DreamNet is significantly more compute-intensive than alternatives like ResNet18. While performance improves, Table 5 shows increased FLOPs, and the paper does not address the cost-benefit trade-off or implications for deployment.

5. Limitation to archtecture. It seems the proposed approach is only applicable to conv net? I'd love to see how the same concept can be applied to transformers.

---

> ### Author Response · Authors · 2025-07-10
> **The Corresponding Response for each comment**
>
> We thank you for your careful reading and constructive feedback. Below, we respond to each major point. We agree with some suggestions and have incorporated them, while for others, we provide clarifications or respectfully disagree, as explained.
>
> ---
>
> 1. **Comment:**
>    *''Ambiguity in dual model design: While the intuition behind simulating sleep is sound, it is unclear why two separate networks are needed. Since dreaming is typically a part of sleep, the decision to separate them into distinct architectures (SleepNet and DreamNet) feels artificial. A more unified framework might be more coherent both biologically and architecturally.''*
>
>     **Our response:**
>     Thank you for this point. We agree that a unified framework would be more biologically faithful and architecturally elegant. Our intention, however, was to empirically isolate the effects of memory consolidation (SleepNet) from those of generative reconstruction (DreamNet). From a technical perspective, DreamNet incorporates additional design elements, such as block ③ in Fig 4 in the paper, which differentiates it from SleepNet and enables a clearer analysis of each component's contribution.
>
> 2. **Comment:**
>    *''Motivation needs to be stronger and more professional: The paper relies on a metaphor, but doesn't sufficiently justify why these biological processes are expected to meaningfully improve learning... I feels the proposed approach seems more of an adhoc method.''*
>
>     **Our response:**
>     We agree that the biological metaphor serves primarily as inspiration. Our primary motivation is to improve classification accuracy through the integration of unsupervised features, guided by the conceptual analogy to sleep and dreaming. This approach is designed to improve the classifiers' performance. We will provide more explanation from this perspective in the next revision of this paper.
>
> 3. **Comment:**
>    *''Contribution feels limited: The building blocks... are not novel. While the composition may be new, more emphasis should be placed on the mechanism by which the combination yields improvements.''*
>
>     **Response:**
>     We agree that the core contribution lies in the mechanism of staged feature integration rather than in the individual network blocks. Our focus is on how this integration yields improvements in accuracy and generalization, as demonstrated in our experiments. We will put in more explanation and emphasis in the next revision of this paper.
>
> 4. **Comment:**
>    *''Computational cost: DreamNet is significantly more compute-intensive than alternatives like ResNet18... the paper does not address the cost-benefit trade-off or implications for deployment.''*
>
>     **Response:**
>     We acknowledge that DreamNet is more computationally intensive. Its design prioritizes accuracy and feature diversity, which may be most valuable in scenarios where performance is critical. We also note that for resource-constrained applications, simpler models or SleepNet alone may be preferable and that is our motivation for providing two types of models.
>
> 5. **Comment:**
>    *''Limitation to architecture: It seems the proposed approach is only applicable to conv net? I'd love to see how the same concept can be applied to transformers.''*
>
>     **Response:**
>     Thank you for your suggestion. We believe the proposed method has strong potential for further development. Our approach relies heavily on the sleep/dream connections, and the experiments based on convolutional architectures are sufficient to demonstrate the effectiveness of these mechanisms. Given the generality of the staged integration principle, we expect that transformer-based models and other architectures would similarly benefit from our approach.
>
> 6. **Comment:**
>    *''Broader impact concerns: The data distribution of dreamnet might be biased... How would you address this problem?''*
>
>     **Response:**
>     We recognize the risk of bias introduced through unsupervised pretraining. To address this, we advocate for careful dataset, monitoring of potential biases to mitigate unintended consequences.

---

> > ### Author Response · Authors · 2025-07-12
> > **Additional comments on broader impact concerns**
> >
> > Thank you again for your thoughtful reviews. I would like to provide additional details regarding the Broader Impact Concerns: *"The data distribution of DreamNet might be biased; an unsupervised approach may lead to biased outcomes. How would you address this problem?"*
> >
> > We fully understand that bias may arise from using pre-trained encoders or autoencoders, especially when they are trained on different datasets with potentially different data distributions. However, from the literature [1], they have theoretically shown that pre-trained models are more likely to introduce beneficial features rather than additional bias. Pre-training provides a strong initialization, which can help the model converge faster and more effectively during downstream training [1].
> >
> > Based on this theory, we believe the pre-trained encoder can help align the learned representations more closely with our specific data distribution, thereby mitigating potential bias. As a result, the bias introduced by the pre-trained encoder is expected to be reduced during training on our own dataset. This is also reflected in our model’s performance, which does not indicate significant bias.
> >
> > Reference:
> > [1] Yao, Y., Yu, B., Gong, C., & Liu, T. (2023). Understanding How Pretraining Regularizes Deep Learning Algorithms. *IEEE Transactions on Neural Networks and Learning Systems, 34*(9), 5828-5840. [https://doi.org/10.1109/TNNLS.2021.3131377](https://doi.org/10.1109/TNNLS.2021.3131377)

---

### Decision · Action_Editor_fVRH · 2025-09-30

**Recommendation:** Reject

**Audience:**

No

**Audience Explanation:**

One reviewer believes there is audience interest (b4EA), while the other two do not, and all reviewers are willing to reject the paper. The AE does not find a strong reason to overturn the consensus.

In summary, the work lacks methodological depth and practical insight beyond its metaphorical framing; it is unlikely to interest TMLR’s research community. The technical contribution and clarity were insufficient to engage TMLR’s core audience. The only reviewer who gave "yes" in audience interest (b4EA) felt the paper would fit better as an exploratory workshop paper rather than a TMLR contribution ("The paper is not ready to be published in TMLR. It feels more like a workshop paper.").

**Claims And Evidence:**

No

**Claims Explanation:**

All three reviewers believe the claims are not well supported, and all reviewers are willing to reject the paper. The AE does not find a strong reason to overturn the consensus.

In summary, the key claim in the paper of biological inspiration and superior performance is not convincingly demonstrated through rigorous experiments, clear metrics, or theoretical justification. And there are also other issues. Specifically,
1. All reviewers have concerns about the claimed connection between the proposed method and biology/neuroscience lacks theoretical or empirical support.
2. Results and metrics were unclear or possibly had issues due to potential errors in FLOP and parameter calculations (FeQp, b4EA).
3. The design of the network relied heavily on metaphors (e.g., “sleep” and “dream”) rather than rigorous analysis (1dVS, FeQp, b4EA).
4. Clarity and writing (1dVS).